# Adjuvant chemotherapy compared with observation in patients with resected biliary tract cancer: A systematic review and meta-analysis of randomized controlled trials

Liying Tian[1], Qian Guo[2], Daidi Fu[3], Xiao Ma[4], Linjun Wang[1]*

**1** Department of Medical, Jinan High-tech East District Hospital, Shandong Healthcare Industry Development Group Co., Ltd, Jinan, Shandong, Peoples' Republic of China, **2** Day Care Unit, Zibo Central Hospital, Shandong University, Zibo, Shandong, Peoples' Republic of China, **3** Department of Oncology, Zibo Central Hospital, Shandong University, Zibo, Shandong, Peoples' Republic of China, **4** Department of Internal Medicine, Zhangqiu People's Hospital, Zhangqiu, Shandong, People' Republic of China

* wanglinjun2023@126.com

## Abstract

### Objectives

Several randomized controlled trials compared adjuvant systemic chemotherapy with observation in patients with resected biliary tract cancer (BTC) have yielded inconsistent outcomes. In order to assess the efficacy of adjuvant therapy in these patients, we conducted this systematic review and meta-analysis.

### Methods

We conducted a thorough search in various databases, which included MEDLINE, EMBASE, Cochrane Central Register of Controlled Trials, ASCO Abstracts, ESMO Abstracts and ClinicalTrials.gov. All relevant randomized controlled trials investigating the adjuvant chemotherapy compared with observation in resected biliary tract cancer were identified. The primary outcome of interest was overall survival (OS), while secondary outcome was relapse-free survival (RFS). Statistical analyses were conducted using Review Manager 5.3. Additionally, publication bias was evaluated using Egger's test in Stata 12.0.

### Results

A total of 5 randomized controlled trials, involving 1406 patients, were included in this analysis. Compared with observation, adjuvant chemotherapy improved RFS [HR 0.84 (0.73-0.96), p=0.01] ($I^2$=0%, p=0.89) but not OS [HR 0.89 (0.77-1.03), p=0.12] ($I^2$=51%, p=0.09) in the entire population after BTC resection. Subgroup analyses revealed that adjuvant chemotherapy did improve both OS [HR 0.76 (0.62-0.93), p=0.009] ($I^2$=7%, p=0.37) and RFS [HR 0.74 (0.58-0.95), p=0.02] ($I^2$=0%, p=0.39) in

**Data availability statement:** All relevant data are within the paper and its Supporting Information files.

**Funding:** The author(s) received no specific funding for this work.

**Competing interests:** The authors have declared that no competing interests exist.

patients with lymph node positivity. Furthermore, patients receiving oral fluoropyrimidine monotherapy showed benefit from the adjuvant therapy, with longer OS [HR 0.78 (0.65-0.94), p=0.009] ($I^2$=2%, p=0.31) and RFS [HR 0.81 (0.68-0.95), p=0.01] ($I^2$=0%, p=0.95).

## Conclusions

To conclude, adjuvant chemotherapy have the potential to offer advantages in patients with resected BTC. Specifically, patients demonstrating positive lymph node status have a higher likelihood of benefiting from adjuvant therapy. Our analysis supports the current standard of care of adjuvant fluoropyrimidine. However, the recommendation of oral fluoropyrimidine monotherapy as the preferred option is not definitive, as it is based on limited studies. Further validation of these outcomes is necessary by conducting extensive randomized controlled trials.

## Introduction

Biliary tract cancer (BTC) encompasses various types of aggressive cancers, including gallbladder cancer, intrahepatic cholangiocarcinoma (iCCA), perihilar cholangiocarcinoma (pCCA) and distal common bile duct cholangiocarcinoma (dCCA) [1]. Cancers originating from the ampulla of Vater, which is the junction of the pancreatic and distal common bile duct, are categorized as biliary tract cancers in Japanese cancer staging systems and are also considered in clinical trials conducted in Japan [2]. BTC is generally divided into 3 types: well-differentiated, moderately-differentiated and poorly-differentiated adenocarcinoma. Rare variants include squamous, adenosquamous, mucinous, signet ring and clear cell types [3].

Approximately 70% of cases of BTC are diagnosed at an advanced stage, making them inoperable [4]. Surgical resection is the only accepted curative approach for patients with resectable disease. However, the recurrence rate remains high, particularly when there is lymph node infiltration or positive surgical margins [5]. The high recurrence rate highlights the need for adjuvant treatment even after radical operation. Historically, there have been limited randomized controlled trials (RCTs) focusing on adjuvant chemotherapy for BTC. As a result, decisions regarding adjuvant chemotherapies are primarily based on data from some retrospective studies and meta-analyses. The crucial meta-analysis, conducted by Horgan et al, which includes only 1 randomized trial, provides strong evidence supporting the use of adjuvant chemotherapy or chemoradiotherapy [6].

In recent years, multiple RCTs have been published, comparing adjuvant systemic chemotherapy with observation in resected BTC. However, these studies have produced conflicting results. To address this, we conducted this systematic review and meta-analysis to evaluate the efficacy of adjuvant chemotherapy versus observation in this patient population.

## Materials and methods

### Search strategy

Literature searches were conducted through MEDLINE, EMBASE, Cochrane Central Register of Controlled Trials, ASCO Abstracts, ESMO Abstracts and ClinicalTrials.gov up to August 2023. The main aim of our search was to specifically find all RCTs in MEDLINE according to OVID format. As a result, the search strategy developed by Cochrane was adopted, which was also suitable in other databases. S2 Table showed the full search strategy used in our analysis. The reference lists of identified review articles were carefully scanned to ensure that eligible studies were covered and included. The searching process had no language restrictions.

### Inclusion and exclusion criteria

Relevant clinical studies were carefully selected based on the following inclusion criteria. (1) The study had to be RCTs about adjuvant chemotherapy compared with observation in resected biliary tract cancer, including gallbladder cancer, iCCA, pCCA, dCCA and ampullary cancer; (2) The study participants had to be adults (≥18 years) with a histologically confirmed biliary tract cancer after curative resection; (3) The study had to report at least one endpoint related to overall survival (OS) or relapse-free survival (RFS); (4) To ensure data consistency, only the most recent publication was included if multiple publications reported data for the same study.

The exclusion criteria were as follows: (1) studies that investigated the adjuvant therapy for periampullary cancer, as it differed significantly from ampullary cancer; (2) studies that investigated neoadjuvant or perioperative therapy for resected BTC; (3) studies that evaluated the combination of radiotherapy, targeted therapy or immunotherapy with chemotherapy; (4) studies with quasi-randomized designs.

Two independent reviewers (Liying Tian and Linjun Wang) conducted the process of study selection using the above criteria. In case of any disagreements, we resolved them by discussing with the third reviewer, Qian Guo.

### Data extraction

Data extraction from all the included studies was carried out independently by Liying Tian and Linjun Wang. In cases of discrepancies and uncertainties, a consensus was achieved with the assistance of the third reviewer, Qian Guo. The primary endpoint for our analysis was OS, defined as the duration from randomization to death resulting from any cause. RFS was the secondary endpoint, defined as the duration from randomization to disease recurrence or death resulting from any cause. We synthesized the hazard ratio (HR) and 95% confidence interval (CI) for OS and RFS following the method described by Jayne F. Tierney et al. [7]. The HRs were collected from the studies incorporated in the analysis or derived from the reported events and the accompanying p value from the log-rank statistics.

### Quality assessment

In order to evaluate the methodological quality of the 5 studies incorporated in this analysis, we implemented the Jadad score approach [8]. The Jadad score assessed 3 crucial aspects: randomization, double blindness, and withdrawals. Each study was assigned 1 point for referring to randomization, double blindness, or withdrawals. Moreover, an additional point was granted if the proper methods of randomization or double blindness were described.

### Statistical analyses

Statistical analyses for the data collected from included studies were conducted using Review Manager 5.3 software. We used HR to analyze time-to-event data (OS and RFS). The $I^2$ statistic, applied in the heterogeneity test, aimed to detect any statistical heterogeneity among the included studies. In cases where $p≥0.10$ or $I^2≤50\%$, it indicated the absence of statistical heterogeneity, and the analysis would be performed using the fixed effect model. Conversely, if $p<0.10$ or

$I^2 > 50\%$, it indicated the presence of statistical heterogeneity, and the random effect model or sensitivity analysis would be applied. Furthermore, we assessed the potential publication bias by performing Egger's test using stata 12.0 software. Statistical significance was defined as a two-sided $p < 0.05$.

## Results

### Study identification

Fig 1 displayed the flow chart outlining the study identification of our analysis. A comprehensive search was conducted, resulting in the identification and screening of 325 articles. After careful consideration, certain articles were excluded for various reasons. Specifically, 157 articles were review articles, 23 articles focused on chemoradiotherapy or radiother-apy, 22 articles had a controlled arm that was not designed for observation, 15 articles were related to immunotherapy or targeted therapy, and 58 articles were case reports or fell under other categories. In addition, the remaining 50 articles underwent further screening for additional details. Out of these, 45 articles were excluded from the study. Among the excluded articles, 23 articles were phase I trials, 12 articles were not designed as controlled trials, 9 articles were retro-spective studies, and 1 article focused specifically on periampullary cancer. Consequently, 5 studies were deemed eligible for inclusion in our analysis, which encompassed the findings of 5 clinical trials as published in the full-text documents [2,9–12]. The total number of included patients in the studies was 1406, with 703 in the chemotherapy arm and 703 in the observation arm.

### Characteristics of included studies

Table 1 summarized the baseline patients' characteristics of the 5 studies. Several differences in patients' characteristics were observed among the included studies. Different from other studies, the study conducted by Sundeep *et al* exclusively enrolled patients with gallbladder cancers, with a majority of them being female [11]. Only the study conducted by Kohei *et al* included patients with ampullary cancer, with a percentage of 17% in both the chemotherapy arm and observation arm [2]. It was important to differentiate between ampullary and periampullary cancer such as extra-biliary or pancreatic cancer because clinical outcomes and therapeutic managements varied widely. As a result, periampullary cancer was not classified as biliary tract cancer. 3 studies had similar-sized R0 resection populations (≥85%) [2,9,10], but the percentage was lower (62%) in the study conducted by John *et al* [12]. All patients in the study conducted by Sundeep *et al* had R0

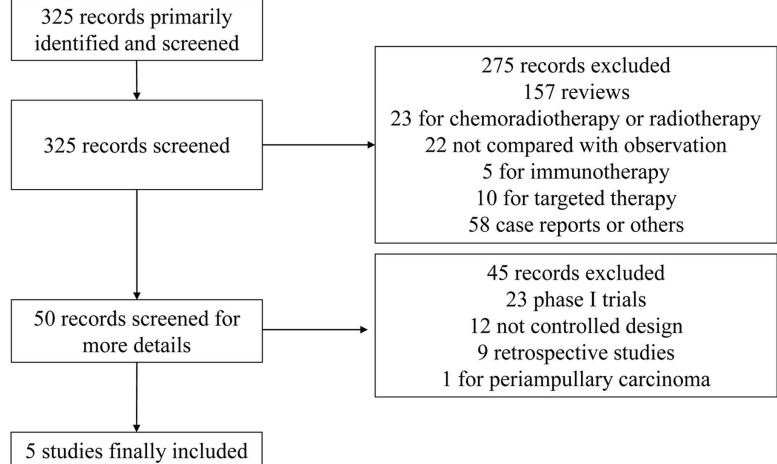

**Fig 1. Flow chart for identification and inclusion of studies for this meta-analysis.**

 

**Table 1. Baseline patients' characteristics of different arms in the included studies.**

| References | Region | Treatment Arms | Patients included | iCCA (%) | pCCA (%) | dCCA (%) | GBC (%) | Ampulla (%) | Male (%) | ECOG 0-1 (%) | Median age (years) | Lymph node negative (%) | R0 (%) |
|---|---|---|---|---|---|---|---|---|---|---|---|---|---|
| T.Ebata et al. 2018 [9] | Japan | Chemo-therapy Observa-tion | 117 108 | 0 0 | 43.6 47.2 | 56.4 52.8 | 0 0 | 0 0 | 65.8 75.9 | 100 100 | NC NC | 64.1 66.7 | 90.6 87.0 |
| Julien et al. 2019 [10] | France | Chemo-therapy Observa-tion | 95 99 | 43 46 | 11 5 | 28 28 | 18 21 | 0 0 | 60 50 | 93 95 | 63 63 | 52 48 | 86 88 |
| Sundeep et al. 2021 [11] | India | Chemo-therapy Observa-tion | 50 50 | 0 0 | 0 0 | 0 0 | 100 100 | 0 0 | 22 14 | NC NC | 50 55 | 50 76 | 100 100 |
| John et al. 2022 [12] | United Kingdom | Chemo-therapy Observa-tion | 223 224 | 19 18 | 29 28 | 34 36 | 18 18 | 0 0 | 50 50 | 97 97 | 62 64 | 52 54 | 62 63 |
| Kohei et al. 2023 [2] | Japan | Chemo-therapy Observa-tion | 218 222 | 12 14 | 21 18 | 36 36 | 14 15 | 17 17 | 74 68 | 100 100 | 68 70 | 61 59 | 86 85 |

iCCA: intrahepatic cholangiocarcinoma. dCCA: distal common bile duct cholangiocarcinoma. pCCA: perihilar cholangiocarcinoma. GBC: gallbladder cancer. ECOG: Eastern Cooperative Oncology Group. NC: no clear. R0: margin-negative resection.

**Table 2. Chemotherapy regimens and endpoints in the included studies.**

| References | Regimens | Interventions | Primary endpoint | Patients completed planned courses (%) | Median follow-up time (months) |
|---|---|---|---|---|---|
| T.Ebata et al. 2018 [9] | Gemcitabine Observation | Arm A: gemcitabine $1000\,mg/m^2$ iv d1,8,15, q4w. Arm B: observation. | OS | 52.1 | 79.4 |
| Julien et al. 2019 [10] | Gemcit-abine + oxa Observation | Arm A: gemcitabine $1000\,mg/m^2$ iv d1, oxa $85\,mg/m^2$ iv d2, q2w. Arm B: observation. | RFS | 74.0 | 46.5 |
| Sundeep et al. 2021 [11] | Gemcit-abine + cis Observation | Arm A: gemcitabine $1000\,mg/m^2$ iv d1,8, cis $80\,mg/m^2$ iv d1, q3w. Arm B: observation. | DFS | 64.0 | 31.5 |
| John et al. 2022 [12] | Capecit-abine Observation | Arm A: capecitabine $1250\,mg/m^2$ po bid d1-14, q3w. Arm B: observation. | OS | NC | 106 |
| Kohei et al. 2023 [2] | S-1 observation | Arm A: S-1 40mg, 50mg or 60mg according to body surface area po bid d1-28, q6w. Arm B: observation. | OS | 68.8 | 45.4 |

oxa: oxaliplatin. cis: cisplatin. S-1: an oral fluoropyrimidine comprising a mixture of tegafur, gimeracil and oteracil potassium. OS: overall survival. RFS: relapse-free survival. DFS: disease-free survival. NC: no clear.

resection [11]. The treatment regimens and endpoints of the studies were shown in **Table 2**. Methodological information, which could potentially introduce bias, was available in **Table 3**. All the 5 included studies were randomized and open-labeled. 4 of the 5 studies were large, phase III and multicenter trial. Among the 5 studies analyzed, 2 of them employed oral fluoropyrimidine monotherapy as adjuvant treatment [2,12], while the remaining 3 studies utilized gemcitabine-based chemotherapy [9–11].

3 studies utilized OS as the primary endpoint [2,9,12], while the other two studies utilized RFS [10] and disease-free survival (DFS) [11] as the primary endpoint respectively. The OS was determined as the period between the randomization and death caused by any reason. Randomization in all 5 studies occurred post-surgery. The RFS was determined as the period between the randomization and disease recurrence or death caused by any reason. The study conducted by Sundeep *et al* used disease-free survival (DFS) as primary endpoint. DFS was defined as the period between the randomization and disease recurrence, which had a significant difference compared with RFS.

The included studies presented subtle variations in the frequency of imaging test. In 2 studies, imaging scans were performed every 3 months for 2 years and every 6 months for the following3 years [10,11]. In 2 other studies, the scans were conducted every 3 months for a duration of 3 years, followed by scans every 6 months for the subsequent 2 years [2,9]. Scans in the last study were conducted every 3 months in the first year, every 6 months in the second year and every 12 months in the remaining 3 years [12]. It was important to highlight that none of the studies explicitly addressed the assessment conducted by the independent review committee regarding the imaging tests.

## Overall survival

All the 5 studies included in the analysis reported data on OS. The pooled HR for OS indicated that there was no improvement in adjuvant chemotherapy compared with observation in resected BTC [HR 0.89 (0.77-1.03), p=0.12], with apparent heterogeneity among studies ($I^2$=51%, p=0.09) (Fig 2). When random-effect model was used, no advantage on OS was observed [HR 0.93 (0.74-1.17), p=0.55] ($I^2$=51%, p=0.09) (S1 Fig).

We conducted subgroup analyses according to the chemotherapy regimens (gemcitabine-based therapy or oral fluoropyrimidine monotherapy), involvement of lymph nodes (lymph nodes negative or positive) and the presence of R0 resection (R0/R1). Adjuvant therapy improved OS in the subgroup receiving oral fluoropyrimidine monotherapy [HR 0.78 (0.65-0.94), p=0.009] ($I^2$=2%, p=0.31). However, it did not show a significant improvement in the subgroup receiving

**Table 3. Methodological details that might lead to bias in the included studies.**

| References | Phase | Random | Blind | Randomization description | Concealment description | Withdraw description | ITT analysis | Multi-center | Jadad score |
|---|---|---|---|---|---|---|---|---|---|
| T.Ebata et al. 2018 [9] | III | Yes | No | Yes | No | Yes | Yes | Yes | 3 |
| Julien et al. 2019 [10] | III | Yes | No | NC | No | Yes | Yes | Yes | 2 |
| Sundeep et al. 2021 [11] | NC | Yes | No | Yes | No | Yes | Yes | No | 3 |
| John et al. 2022 [12] | III | Yes | No | Yes | No | Yes | Yes | Yes | 3 |
| Kohei et al. 2023 [2] | III | Yes | No | Yes | No | Yes | Yes | Yes | 3 |

NC: no clear. ITT: intend-to-treat

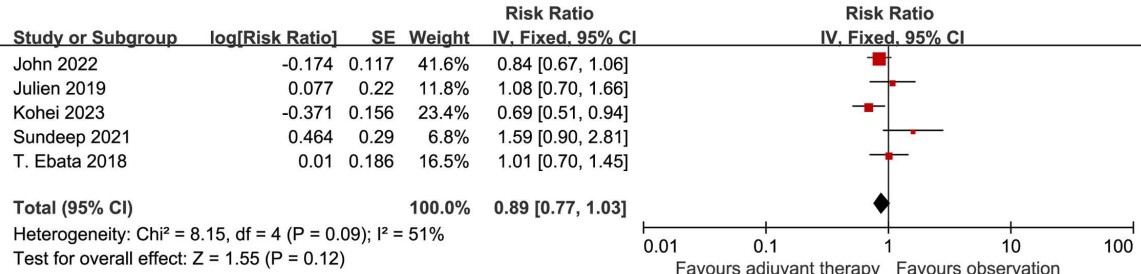

**Fig 2. Comparison of OS between adjuvant chemotherapy and observation.** SE: standard error. CI: Confidence interval. IV: Inverse variance.

gemcitabine-based chemotherapy [HR 1.13 (0.88-1.45), p=0.35] ($I^2$=0%, p=0.41) (S2 Fig). 4 studies reported the subgroup analysis of OS related to lymph nodes status and presence of R0 resection [2,9–11]. All patients included in the study conducted by Sundeep *et al* underwent R0 resection [11]. The most recent publication related to the study conducted by John *et al* did not provide the mentioned subgroup analysis [12]. We extracted the necessary data from their earlier publication in 2019 [13]. Compared with observation, adjuvant chemotherapy showed benefits on OS in the subgroup of lymph nodes positive [HR 0.76 (0.62-0.93), p=0.009] ($I^2$=7%, p=0.37), but not in the subgroup of lymph nodes negative [HR 0.95 (0.75-1.20), p=0.68] ($I^2$=0%, p=0.76) (S3 Fig). Adjuvant chemotherapy did not show benefit on OS in the both subgroups of R0 resection [HR 0.86 (0.72-1.03), p=0.09] ($I^2$=54%, p=0.07) and R1 resection [HR 0.97 (0.73-1.29), p=0.84] ($I^2$=0%, p=0.91) (S4 Fig). Heterogeneity was found in the subgroup analysis of R0 resection. But final result did not change when a random-effect model was applied [HR 0.90 (0.69-1.18), p=0.45] ($I^2$=54%, p=0.07) (S5 Fig).

## Relapse-free survival

4 of the 5 included studies reported the data on RFS [2,9,10,12]. The pooled HR for RFS showed advantage in adjuvant chemotherapy compared with observation in resected BTC [HR 0.84 (0.73-0.96), p=0.01], without apparent heterogeneity among studies ($I^2$=0%, p=0.89) (Fig 3).

Adjuvant therapy improved RFS in the subgroup receiving oral fluoropyrimidine monotherapy [HR 0.81 (0.68-0.95), p=0.01] ($I^2$=0%, p=0.95), but not in the subgroup receiving gemcitabine-based chemotherapy [HR 0.90 (0.71-1.16), p=0.43] ($I^2$=0%, p=0.83) (S6 Fig). 3 studies reported the subgroup analysis of RFS related to lymph nodes status and presence of R0 resection [2,9,10]. Compared with observation, adjuvant chemotherapy improved RFS in the subgroup of lymph nodes positive [HR 0.74 (0.58-0.95), p=0.02] ($I^2$=0%, p=0.39), but not in the subgroup of lymph nodes negative [HR 0.95 (0.73-1.24), p=0.70] ($I^2$=0%, p=0.93) (S7 Fig). Adjuvant chemotherapy did not bring benefit on RFS in the both subgroups of R0 resection [HR 0.83 (0.68-1.01), p=0.06] ($I^2$=0%, p=0.80) and R1 resection [HR 0.94 (0.61-1.44), p=0.78] ($I^2$=0%, p=0.78) (S8 Fig).

## Sensitivity analysis

It was important to note that there were limited studies included in the subgroup analysis. Upon conducting subgroup analysis of RFS based on lymph node involvement (negative or positive) and R0 resection status (R0/R1), it was observed that the overall benefit appeared to be due to the influence of a dominant study conducted by Kohei *et al*, which carried significantly more weight than others (S7 and S8 Figs). To evaluate the impact of individual study on the meta-analysis, we systematically excluded one study at a time and examined its effect on the overall summary estimate. When excluding the study conducted by Kohei *et al*, the analysis outcome of RFS based on lymph node positive involvement changed [HR 0.88 (0.61-1.26), p=0.48] ($I^2$=0%, p=0.66) (S9 Fig). No other individual study had a significant impact on the overall outcome or the heterogeneity of the subgroup analysis mentioned.

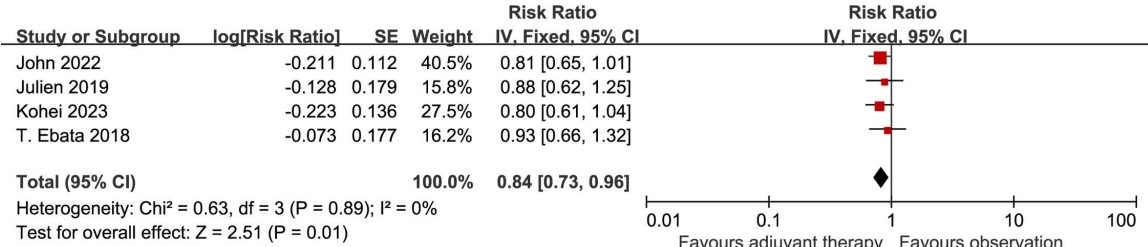

**Fig 3. Comparison of RFS between adjuvant chemotherapy and observation.** SE: standard error. CI: Confidence interval. IV: Inverse variance.

## Publication bias

To prevent the occurrence of publication bias, we implemented a meticulously crafted search approach to identify pertinent studies, effectively minimizing any potential issues. Publication bias was assessed using funnel plots. Egger's test provided statistical evidence of bias. Our analysis revealed no apparent publication bias, as shown in the funnel plots (Fig 4). There was no significant publication bias for OS (p=0.122) and RFS (p=0.138) analyzed by Egger's test.

## Discussion

Over the past decade, progress in the therapeutic scope of BTC have been slow. Neoadjuvant therapy is supposed to increase R0 resection rate and provide survival benefits in selected group of patients [14]. Moreover, adjuvant treatments after resection have shown promise in improving patients' outcomes. However, due to the relative rarity and significant heterogeneity of anatomical subtypes of BTC, limited number of randomized prospective clinical trials have been conducted. Several studies have yielded inconsistent results.

Our pooled analysis indicated that adjuvant chemotherapy improved RFS in the entire population after BTC resection. Additionally, it improved both OS and RFS in patients with lymph node positivity and those receiving oral fluoropyrimidine monotherapy. According to Kohei *et al*, the median RFS was 5.3 years in the S-1 group and 3.5 years in the observation group, resulting in a 3-year OS rate advantage of 77.1% (95% CI, 70.9 to 82.1%) compared to 67.6% (61.0 to 73.3%) [2]. Similarly, in the study conducted by John *et al*, median RFS was 24.3 (18.6 to 34.6) months for capecitabine and 17.4 (11.8 to 23.0) months for surveillance. Additionally, the median OS was 49.6 (35.1 to 59.1) months for capecitabine and 36.1 (29.7 to 44.2) months for the observation group [12]. The possible explanations for the observed benefits of adjuvant therapy when using oral fluoropyrimidine might be its well-tolerated nature and lower toxicities. For example, in the study conducted by Kohei *et al* [2], S-1, known for its well-tolerated nature, had a high treatment completion rate compared to gemcitabine or capecitabine. Furthermore, gemcitabine-based chemotherapy regimens were associated with more frequent haematological toxicities, such as neutropenia and thrombocytopenia. Consequently, this resulted in a higher rate of dose reduction during the treatment procedure [9–11]. As a result, it is important to fully consider the completion level and treatment compliance of patients when deciding on postoperative adjuvant therapy in resected BTC. Nevertheless,

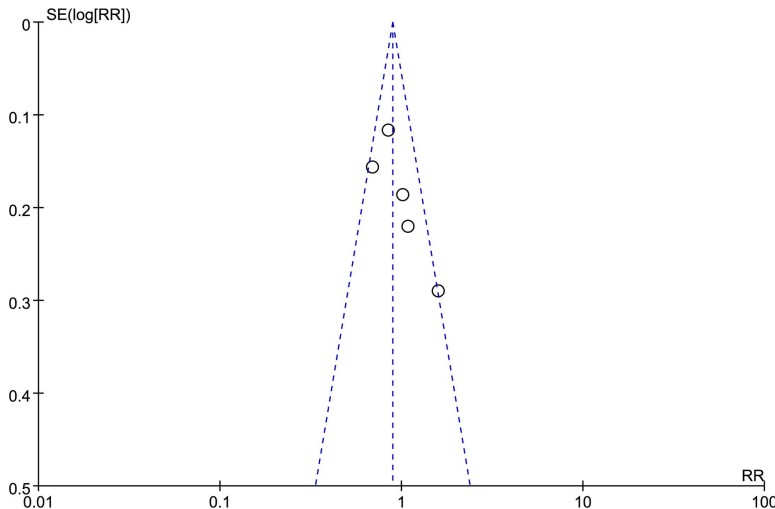

**Fig 4. Publication bias shown in funnel plots.**

the subgroup analysis only included two studies that utilized oral fluoropyrimidine monotherapy regimens, highlighting the necessity for additional RCTs for a definitive conclusion.

In our analysis, adjuvant chemotherapy improved both OS and RFS in the subgroup of patients with lymph node involvement. However, we did not observe any advantage in the subgroup analyses of patients without lymph node involvement, those who had undergone R0 resection, or those who had undergone R1 resection. The role of pathology in predicting outcomes in patients with resected BTC have been highlighted, particularly by considering well-established prognostic indicators such as lymph node involvement [15]. Lymph node involvement is strongly linked to an elevated risk of early relapse in patients with resected BTC. Therefore, it is crucial to perform lymphadenectomy as a standard procedure in patients undergoing liver resection, even if they do not have clinically detectable lymph node involvement [16]. This might explain why adjuvant chemotherapy have shown benefits in populations with positive lymph nodes in our analysis.

However, there is no validation for the use of predictive or prognostic biomarkers in clinical practice. The exploration in this area have been limited to a few studies conducted in small groups of individuals, and the results obtained have been inconclusive. The serum biomarker that continued to be widely used is CA19–9 [17,18]. In our analysis, we were unable to extract the necessary data and perform an effective analysis due to the limited number of studies that included subgroup analysis based on CA19–9 concentration. And the included studies used different boundary points. To establish predictors of treatment response that can be applied in a clinical setting, gain insight into both primary and acquired resistance, and identify suitable future treatment options in adjuvant therapy, it is essential to conduct large-scale prospective studies with the uniform subgroup set of prognostic biomarkers.

The immunogenicity characteristics of BTC involves the up-regulation of programmed cell death ligand 1 (PD-L1) and DNA mismatch repair (dMMR)/high microsatellite instability (MSI-H) [19]. The MOUSEION-03 meta-analysis revealed that the use of immunotherapy might significantly increase the likelihood of achieving complete response in various solid tumors compared to control treatments [20]. In the phase II KEYNOTE-158 study, pembrolizumab showed favorable outcomes in patients with previously treated dMMR/MSI-H advanced BTC. Subgroup analysis showed that the ORR of the BTC MSI-H subgroup was 40.9% [21]. In the phase III KEYNOTE-966 study, the combination of pembrolizumab with gemcitabine and cisplatin improved OS compared to chemotherapy alone (12.7 vs 10.9 months) [22]. These findings ultimately lead to the approval of immunotherapy by the United States Food and Drug Administration (FDA) for patients with advanced BTC who have experienced treatment failure after systemic treatment. Moreover, immunotherapy have shown promise in patients with resectable lung or esophageal cancer [23,24], suggesting its potential advantages in adjuvant treatment of BTC. Furthermore, significant advancements have been made in understanding the molecular features of BTC. These findings have laid the foundation for the development of targeted treatments, which have been evaluated both as monotherapy and in combination with chemotherapy [25]. Additional clinical trials should be necessary to assess the efficacy of incorporating immune checkpoint inhibitors or targeted drugs into chemotherapy for individuals with resected BTC.

There were several limitations existed in our analysis. Firstly, there were substantial variations in patients' characteristics among different studies for each subtype of BTC. Considering the diverse etiology, molecular characteristics, prognosis, and natural progression in different anatomical subcategories of BTC, it was reasonable to assume that these variables might have bring significant heterogeneity among studies. Secondly, the small number of included studies limited the statistical power of the analysis, especially in the subgroup analysis. Sensitivity analysis showed that the study conducted by Kohei *et al* had a dominant effect in the subgroup analysis of RFS based on lymph node involvement and R0 resection status. Future clinical trials of adjuvant chemotherapy should target patients with high-risk features, such as positive lymph node status post-resection, to gather additional data that reinforces our conclusion. Thirdly, there was inconsistency in the primary endpoint used across the studies. OS was used in 3 studies, whereas DFS or RFS was used in 2 studies. It should be noted that the follow-up periods might have differed between studies using OS and those using DFS or RFS, potentially resulting

in varying levels of data maturity among studies. Fourthly, it was important to note that all the studies included in the analysis had a Jadad score of 3, indicating relatively low quality. This lower quality was primarily due to the open-label design used. Due to ethical considerations, achieving a double-blind trial design for postoperative adjuvant therapy was challenging. Fifthly, three of the studies analyzed involved Asian patients [2,9,11], while the remaining two studies focused on European patients [10,12]. Research has indicated potential variations in the pharmacokinetics and toxicity of chemotherapy drugs across different racial backgrounds. For instance, differences in the toxicity profile of S-1 were observed between European and North American patients, specifically in relation to the incidence of diarrhoea, resulting in a need for dose adjustment [26,27]. Therefore, large-sample, international and multi-center RCTs are needed. Finally, due to the lack of adequate data in eligible studies, we were unable to uncover any potential advantages in terms of OS and RFS by employing the adjuvant treatment in various patient groups with differing detailed stages, ages, general conditions, and so on.

In conclusion, adjuvant chemotherapy can provide benefits in patients with resected BTC. Specifically, patients with lymph node positive status are more likely to benefit from adjuvant therapy. The recommendation of oral fluoropyrimidine monotherapy as the preferred option is not conclusive, as it relies on limited studies. Further validation of these results is necessary through additional large-scale randomized controlled trials.

## Supporting information

**S1 Table. PRISMA checklist.**
(DOCX)

**S2 Table. The MEDLINE (OVID format) search strategy used in this meta-analysis.**
(DOCX)

**S1 Appendix. PRISMA flow diagram.**
(DOCX)

**S1 Fig. Comparison of OS between adjuvant chemotherapy and observation using random-effect model.** SE: Standard error. CI: Confidence interval. IV: Inverse variance.
(TIF)

**S2 Fig. Comparison of OS between adjuvant chemotherapy and observation in the different subgroups based on chemotherapy regimens.** SE: Standard error. CI: Confidence interval. IV: Inverse variance.
(TIF)

**S3 Fig. Comparison of OS between adjuvant chemotherapy and observation in the different subgroups based on lymph nodes status.** SE: Standard error. CI: Confidence interval. IV: Inverse variance.
(TIF)

**S4 Fig. Comparison of OS between adjuvant chemotherapy and observation in the different subgroups based on R0/R1 resection.** SE: Standard error. CI: Confidence interval. IV: Inverse variance.
(TIF)

**S5 Fig. Comparison of OS between adjuvant chemotherapy and observation in the different subgroups based on R0/R1 resection using random-effect model.** SE: Standard error. CI: Confidence interval. IV: Inverse variance.
(TIF)

**S6 Fig. Comparison of RFS between adjuvant chemotherapy and observation in the different subgroups based on chemotherapy regimens.** SE: Standard error. CI: Confidence interval. IV: Inverse variance.
(TIF)

**S7 Fig. Comparison of RFS between adjuvant chemotherapy and observation in the different subgroups based on lymph nodes status.** SE: Standard error. CI: Confidence interval. IV: Inverse variance.
(TIF)

**S8 Fig. Comparison of RFS between adjuvant chemotherapy and observation in the different subgroups based on R0/R1 resection.** SE: Standard error. CI: Confidence interval. IV: Inverse variance.
(TIF)

**S9 Fig. Comparison of RFS between adjuvant chemotherapy and observation in the different subgroups based on lymph nodes status excluding Kohei *et al*.** SE: Standard error. CI: Confidence interval. IV: Inverse variance.
(TIF)

## Acknowledgments

We would like to express our gratitude to Wei Tian for his valuable advice on the search strategy.

## Author contributions

**Data curation:** Liying Tian, Linjun Wang.

**Formal analysis:** Daidi Fu, Linjun Wang.

**Investigation:** Linjun Wang.

**Methodology:** Qian Guo, Daidi Fu, Linjun Wang.

**Resources:** Qian Guo.

**Software:** Liying Tian, Qian Guo, Xiao Ma.

**Supervision:** Liying Tian, Linjun Wang.

**Validation:** Xiao Ma.

**Writing – original draft:** Liying Tian, Daidi Fu, Xiao Ma.

**Writing – review & editing:** Linjun Wang.

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
