## [Decision Letter · Decision Letter 0]

13 Oct 2023

PONE-D-23-30589Adjuvant chemotherapy compared with observation in patients with resected biliary tract cancer: a systematic review and meta-analysis of randomized controlled trialsPLOS ONE

Dear Dr. Wang,

Thank you for submitting your manuscript to PLOS ONE. After careful consideration, we feel that it has merit but does not fully meet PLOS ONE’s publication criteria as it currently stands. Therefore, we invite you to submit a revised version of the manuscript that addresses the points raised during the review process.

We look forward to receiving your revised manuscript.

Kind regards,

Alessandro Rizzo

Academic Editor

PLOS ONE

Journal Requirements:

2. Please include a caption for figure 1.

Reviewers' comments:

Reviewer's Responses to Questions

**Comments to the Author**

1. Is the manuscript technically sound, and do the data support the conclusions?

Reviewer #1: Yes

Reviewer #2: Partly

2. Has the statistical analysis been performed appropriately and rigorously? 

Reviewer #1: I Don't Know

Reviewer #2: Yes

3. Have the authors made all data underlying the findings in their manuscript fully available?

Reviewer #1: Yes

Reviewer #2: Yes

4. Is the manuscript presented in an intelligible fashion and written in standard English?

Reviewer #1: Yes

Reviewer #2: No

5. Review Comments to the Author

Reviewer #1: Adjuvant chemotherapy compared with observation in patients with resected biliary tract cancer: a systematic review and meta-analysis of randomized controlled trials.

Conclusions: To conclude, adjuvant chemotherapy had the potential to offer advantages in patients with resected BTC.

major revision

How long has adjuvant chemotherapy extended the survival period?

Reviewer #2: The study assesses a current, timely topic in biliary tract cancer. An impressive number of patients and studies have been included.

A linguistic revision is required.

We believe this article is suitable for publication in the journal although some revisions are needed. The main strengths of this paper are that it addresses an interesting and very timely question and provides a clear answer, with some limitations. Certainly, the authors should better highlight the limitations of the current paper.

- The background of the changing scenario of medical treatment in BTC should be better discussed, and some recent papers regarding this topic should be included ( PMID: 36633661; PMID: 33592561; PMID: 33756174 ; PMID: 35031442 ).

6. PLOS authors have the option to publish the peer review history of their article (what does this mean? ). If published, this will include your full peer review and any attached files.

**Do you want your identity to be public for this peer review?** For information about this choice, including consent withdrawal, please see our Privacy Policy .

Reviewer #1: **Yes: ** Wei Liu

Reviewer #2: No

---

## [Author Response · Author response to Decision Letter 0]

20 Nov 2023

December 16, 2023

Reply for manuscript “Adjuvant chemotherapy compared with observation in patients with resected biliary tract cancer: a systematic review and meta-analysis of randomized controlled trials”

Reply to editor:

Dear editor:

Thank you for providing us with the opportunity to submit the revised manuscript. We would like to express our sincere gratitude to all the reviewers for their constructive comments. Based on these helpful comments, we have extensively revised the manuscript by correcting mistakes and supplementing the required materials to enhance its quality. Additionally, we have ensured that the manuscript adheres to PLOS ONE's style requirements.

Journal Requirements:

Answer: We have carefully reviewed our manuscript to ensure that it meets PLOS ONE's style requirements.

2. Please include a caption for figure 1.

Answer: We have added the caption for figure 1. Line 90-92.

Answer: We have added captions for Supporting information in the end of our manuscript. Line 340-342.

Reply to reviewers:

Dear reviewer:

We greatly appreciate your time and helpful comments on our manuscript. These comments are valuable and greatly help our manuscript. We have considered each comment and replied individually and marked the line number in the revised manuscript where we track the change. We hope you are more satisfied with the revised version. We would be happy to make further revisions to the manuscript.

Reply to reviewer 1:

Reviewer #1: Adjuvant chemotherapy compared with observation in patients with resected biliary tract cancer: a systematic review and meta-analysis of randomized controlled trials.

Conclusions: To conclude, adjuvant chemotherapy had the potential to offer advantages in patients with resected BTC. How long has adjuvant chemotherapy extended the survival period?

Answer: We have added the survival benefit scenarios to the Discussion section. Line 258-262.

Reply to reviewer 2:

Reviewer #2: The study assesses a current, timely topic in biliary tract cancer. An impressive number of patients and studies have been included.

A linguistic revision is required.

Answer: The manuscript has been linguistically polished, especially in the background and discussion sections.

We believe this article is suitable for publication in the journal although some revisions are needed. The main strengths of this paper are that it addresses an interesting and very timely question and provides a clear answer, with some limitations. Certainly, the authors should better highlight the limitations of the current paper.

Answer: We have enriched the limitation part of our manuscript by including a more comprehensive discussion. Line 326-331.

- The background of the changing scenario of medical treatment in BTC should be better discussed, and some recent papers regarding this topic should be included ( PMID: 36633661; PMID: 33592561; PMID: 33756174 ; PMID: 35031442 ).

Answer: We have enriched our discussion part of the manuscript based on the recent papers, and added the papers (PMID: 36633661; PMID: 33592561; PMID: 33756174) in the reference. Line 250-251. Line 298-300. Line 312-314.

---

## [Decision Letter · Decision Letter 1]

19 Jan 2024

PONE-D-23-30589R1Adjuvant chemotherapy compared with observation in patients with resected biliary tract cancer: a systematic review and meta-analysis of randomized controlled trialsPLOS ONE

Dear Dr. Wang,

Thank you for submitting your manuscript to PLOS ONE. After careful consideration, we feel that it has merit but does not fully meet PLOS ONE’s publication criteria as it currently stands. Therefore, we invite you to submit a revised version of the manuscript that addresses the points raised during the review process.

**ACADEMIC EDITOR: **Please reconsider the conclusions drawn about monotherapy using oral fluoropyrimidine regimens and ensure they are supported by your meta-analysis findings.Please reduce the number of figures to the most essential ones, relegating the rest to supplementary material.Please ensure the manuscript is reviewed by a native English speaker for language correctness.Please address the influence of a dominant study in your subgroup analysis and discuss the implications of this in your discussion section.==============================

We look forward to receiving your revised manuscript.

Kind regards,

Jincheng Wang

Academic Editor

PLOS ONE

Additional Editor Comments:

Please reconsider the conclusions drawn about monotherapy using oral fluoropyrimidine regimens and ensure they are supported by your meta-analysis findings.

Please reduce the number of figures to the most essential ones, relegating the rest to supplementary material.

Please ensure the manuscript is reviewed by a native English speaker for language correctness.

Please address the influence of a dominant study in your subgroup analysis and discuss the implications of this in your discussion section.

Reviewers' comments:

Reviewer's Responses to Questions

**Comments to the Author**

1. If the authors have adequately addressed your comments raised in a previous round of review and you feel that this manuscript is now acceptable for publication, you may indicate that here to bypass the “Comments to the Author” section, enter your conflict of interest statement in the “Confidential to Editor” section, and submit your "Accept" recommendation.

Reviewer #1: All comments have been addressed

Reviewer #3: (No Response)

Reviewer #4: (No Response)

Reviewer #5: All comments have been addressed

Reviewer #6: (No Response)

2. Is the manuscript technically sound, and do the data support the conclusions?

Reviewer #1: Yes

Reviewer #3: Yes

Reviewer #4: Yes

Reviewer #5: Yes

Reviewer #6: Yes

3. Has the statistical analysis been performed appropriately and rigorously? 

Reviewer #1: I Don't Know

Reviewer #3: Yes

Reviewer #4: Yes

Reviewer #5: N/A

Reviewer #6: Yes

4. Have the authors made all data underlying the findings in their manuscript fully available?

Reviewer #1: Yes

Reviewer #3: Yes

Reviewer #4: Yes

Reviewer #5: No

Reviewer #6: Yes

5. Is the manuscript presented in an intelligible fashion and written in standard English?

Reviewer #1: Yes

Reviewer #3: Yes

Reviewer #4: Yes

Reviewer #5: Yes

Reviewer #6: No

6. Review Comments to the Author

Reviewer #1: Adjuvant chemotherapy compared with observation in patients with resected biliary tract cancer: a systematic review and meta-analysis of randomized controlled trials

Conclusions: To conclude, adjuvant chemotherapy had the potential to offer advantages in patients with resected BTC. Specifically, patients demonstrating positive lymph node status had a higher likelihood of benefiting from adjuvant therapy. The administration of monotherapy using oral fluoropyrimidine regimens should be regarded as the primary suggestion. Nonetheless, further validation of these outcomes was necessary by conducting extensive randomized controlled trials.

The author provided a good explanation of the problem

Reviewer #3: In their manuscript entitled “Adjuvant chemotherapy compared with observation in patients with resected biliary tract cancer: a systematic review and meta-analysis of randomized controlled trials”, Tian et al. aimed to perform a systematic review and meta-analysis to examine the impact of adjuvant chemotherapy on survival of patients with resected biliary tract cancers. The individual trials reported that adjuvant fluoropyrimidine improves survival (but gem-based chemotherapy does not, yet), and this is confirmed by this meta-analysis. In fact, since the included trials are limited and fairly heterogeneous, I would probably not attempt to synthesize the results using meta-analysis. That said, the review/analysis seems well-done and the subgroup analyses are appropriate. Additional comments:

1) In their conclusions, the authors note that “The administration of monotherapy using oral fluoropyrimidine regimens should be regarded as the primary suggestion” – this is supported by individual trials, but not the meta-analysis (since they did not compare different types of chemotherapy). I would change the wording in this regard (could be something like “our analysis supports the current standard of care of adjuvant fluoropyrimidine”, or something similar).

2) Some surgeons may argue that ampullary cancers do not belong to the BTC group (rather pancreas) – I would recommend the authors specify how they handled ampullary cancers (included or excluded in each study).

3) The presentation of the study is very confusing, with tables/figures/figure legends in the middle of the manuscript, not sure if this was intentional or a software malfunction.

4) All trials were relatively low quality – this should be included in the limitations.

5) I’d recommend that the authors convert their current table 1 (search terms) to a supplemental table. If available, I would add additional info to tables 2-3, including actual numbers or percentages for tumor location among the studies, type of surgery, margin status, pertinent inclusion/exclusion criteria (if applicable), how OS was defined (randomized before or after surgery), their followup period, how many patients completed the planned chemo course, chemo dose density etc . Some extra info/examples are included in these papers: 31229583, 33787841.

6) In my opinion, there are too many figures and the reader gets lost (especially since the figure legends are not readily available). I would recommend the authors include up to 4-5 figures (choose the most important ones) and submit the rest as supplemental material.

7) The manuscript needs to be reviewed by a native English speaker for syntax /grammar (i.e. use of past tense instead of present tense etc).

8) Some of the DOI links in the references are not working, please double check and update accordingly.

Reviewer #4: The design and statistical analyses of this meta-analysis are reasonable, and the search process appears to have been conducted effectively. However, I would like to point out two issues that the author should address.

Firstly, it has come to my attention that there is already one paper titled 'meta-analysis in resected biliary tract cancer (the sixth paper in the references). Therefore, it is important for the author to clarify what new contributions this paper will make to the existing literature.

Secondly, in the 'Revised Manuscript with Track Change', there should be a "blank space" inserted in front of '3 years' on Line 173.

Reviewer #5: The manuscript titled "Adjuvant chemotherapy compared with observation in patients with resected biliary tract cancer: a systematic review and meta-analysis of randomized controlled trials" provides a comprehensive evaluation of the efficacy of adjuvant chemotherapy in patients with resected biliary tract cancer (BTC). Here are critical points and observations:

### Title and Author Information

- The title clearly reflects the content and purpose of the study.

- The authors' affiliations are listed, indicating a multidisciplinary and collaborative effort.

- The corresponding author and contact information are provided.

### Abstract

- The abstract succinctly summarizes the study's objectives, methods, results, and conclusions, which provides a clear overview for readers.

- It includes key findings, such as the impact of adjuvant chemotherapy on relapse-free survival (RFS) and overall survival (OS) in patients with lymph node-positive status.

### Introduction

- The introduction provides necessary background information and context for the study.

- It explains the significance of the clinical issue and the rationale behind conducting the study, which justifies the need for this systematic review and meta-analysis.

### Methods

- Comprehensive databases were searched, with a clear outline of the search strategy, which indicates thoroughness.

- The inclusion and exclusion criteria are well-defined, ensuring the selection of relevant studies.

- The methodological approach, including data extraction and quality assessment, is meticulously detailed.

- The statistical analyses are described, with software and methods mentioned, enhancing reproducibility.

### Results

- The results are thoroughly detailed, providing information on study selection, patient characteristics, and outcomes.

- The heterogeneity among studies has been explained, contributing to transparency.

- Subgroup analyses and their implications are particularly well-detailed, which helps understand nuances in the data.

### Discussion

- The discussion contextualizes the findings within the existing literature.

- Limitations are openly discussed, providing a balanced view of the study's implications.

- Suggestions for future research are valuable and point towards potential advancements in the field.

### General Observations

- The study presents a well-organized flow of information from the background to the implications of the findings.

- References are current and relevant, indicating comprehensive literature engagement.

- Tables and figures are likely informative (though not viewable in this format), providing crucial visualization of data.

- The writing style is scholarly and seems to maintain clinical research formalities.

### Specific Considerations for Improvement

- There may be a need for a more in-depth exploration of bias within included studies, especially given the open-label nature of some trials.

- Consider discussing the geographic distribution of the included studies and its potential impact on the findings.

- Ensure a thorough proofread to catch any typographical errors and inconsistencies in formatting (not evident from the provided text).

### Final Thoughts

- The study seems to add valuable information about the efficacy of adjuvant chemotherapy for BTC.

- It highlights the importance of patient selection (lymph node-positive patients) for adjuvant chemotherapy.

- Further research, as recommended by the authors, is imperative to solidify the findings and potentially influence clinical practice.

By addressing these considerations, the paper could make an even more robust contribution to the field and serve as a credible source of information for clinical decision-making in the treatment of resected BTC.

Reviewer #6: Thank you for the opportunity to review your manuscript which evaluated the benefit of adjuvant chemotherapy in patients with post-resected biliary tract cancer. Your meta-analysis included 5 studies with different patient populations, chemotherapy regimens, and different outcome measures. Overall you found significant benefits for adjuvant chemotherapy in patients without R0 resection and those with node-positive disease. Due to the limited evidence in this field, your study makes a significant contribution.

I have a few minor recommendations:

1. The are some grammatical errors that need to be corrected

2. There were few studies in the subgroup analysis, and the overall benefit appeared to be due to the influence of a dominant study (Kohei 2023- Benefit of lymph node-positive disease). You need to be sensitive to this and address it in your discussion.

7. PLOS authors have the option to publish the peer review history of their article (what does this mean? ). If published, this will include your full peer review and any attached files.

**Do you want your identity to be public for this peer review?** For information about this choice, including consent withdrawal, please see our Privacy Policy .

Reviewer #1: **Yes: ** Wei Liu

Reviewer #3: No

Reviewer #4: **Yes: ** Rui Wang

Reviewer #5: **Yes: ** AHMAD O. KHALIFA

Reviewer #6: No

---

## [Author Response · Author response to Decision Letter 1]

4 Mar 2024

March 04, 2024

Reply for manuscript “Adjuvant chemotherapy compared with observation in patients with resected biliary tract cancer: a systematic review and meta-analysis of randomized controlled trials”

Reply to editor:

Dear editor:

Thank you for providing us with the opportunity to submit the revised manuscript. We would like to express our sincere gratitude to all the reviewers for their constructive comments. Based on these helpful comments, we have extensively revised the manuscript by correcting mistakes and supplementing the required materials to enhance its quality.

Additional Editor Comments:

1.Please reconsider the conclusions drawn about monotherapy using oral fluoropyrimidine regimens and ensure they are supported by your meta-analysis findings.

Answer: We revised the conclusion about oral fluoropyrimidine regimens. We could not make a definitive conclusion that oral fluoropyrimidine monotherapy is superior to combination chemotherapy. Line 51-53, Line 342-343.

2.Please reduce the number of figures to the most essential ones, relegating the rest to supplementary material.

Answer: We reduced the number of figures and relegated the rest to the supplementary materials.

3.Please ensure the manuscript is reviewed by a native English speaker for language correctness.

Answer: Our manuscript was polished by a native English speaker to enhance its quality.

4.Please address the influence of a dominant study in your subgroup analysis and discuss the implications of this in your discussion section.

Answer: We added sensitivity analysis in the manuscript, and discussed the dominant study in the limitation part. Line 237-246, Line 318-322.

Reply to reviewers:

Reviewer #1: Adjuvant chemotherapy compared with observation in patients with resected biliary tract cancer: a systematic review and meta-analysis of randomized controlled trials

Conclusions: To conclude, adjuvant chemotherapy had the potential to offer advantages in patients with resected BTC. Specifically, patients demonstrating positive lymph node status had a higher likelihood of benefiting from adjuvant therapy. The administration of monotherapy using oral fluoropyrimidine regimens should be regarded as the primary suggestion. Nonetheless, further validation of these outcomes was necessary by conducting extensive randomized controlled trials.

The author provided a good explanation of the problem

Reviewer #3: In their manuscript entitled “Adjuvant chemotherapy compared with observation in patients with resected biliary tract cancer: a systematic review and meta-analysis of randomized controlled trials”, Tian et al. aimed to perform a systematic review and meta-analysis to examine the impact of adjuvant chemotherapy on survival of patients with resected biliary tract cancers. The individual trials reported that adjuvant fluoropyrimidine improves survival (but gem-based chemotherapy does not, yet), and this is confirmed by this meta-analysis. In fact, since the included trials are limited and fairly heterogeneous, I would probably not attempt to synthesize the results using meta-analysis. That said, the review/analysis seems well-done and the subgroup analyses are appropriate. Additional comments:

1) In their conclusions, the authors note that “The administration of monotherapy using oral fluoropyrimidine regimens should be regarded as the primary suggestion” – this is supported by individual trials, but not the meta-analysis (since they did not compare different types of chemotherapy). I would change the wording in this regard (could be something like “our analysis supports the current standard of care of adjuvant fluoropyrimidine”, or something similar).

Answer: We changed the conclusion part of our manuscript. “Our analysis supports the current standard of care of adjuvant fluoropyrimidine. However, the recommendation of oral fluoropyrimidine monotherapy as the preferred option is not definitive, as it is based on limited studies”. Line 51-53, Line 342-343.

2) Some surgeons may argue that ampullary cancers do not belong to the BTC group (rather pancreas) – I would recommend the authors specify how they handled ampullary cancers (included or excluded in each study).

Answer: Only the study conducted by Kohei et al included patients with ampullary cancer, with a percentage of 17% in both the chemotherapy arm and observation arm. Cancers originating from the ampulla of Vater were categorized as biliary tract cancers in Japanese cancer staging systems and were also considered in clinical trials conducted in Japan. So our inclusion criteria demonstrated that ampullary cancer should be included. Line 59-61, Line 149-150.

3) The presentation of the study is very confusing, with tables/figures/figure legends in the middle of the manuscript, not sure if this was intentional or a software malfunction.

Answer: The presentation of the tables and figures was following the guidelines for manuscript of PLOS ONE.

4) All trials were relatively low quality – this should be included in the limitations.

Answer: We added the reason of the lower quality of studies in the discussion part. Line 326-328.

5) I’d recommend that the authors convert their current table 1 (search terms) to a supplemental table. If available, I would add additional info to tables 2-3, including actual numbers or percentages for tumor location among the studies, type of surgery, margin status, pertinent inclusion/exclusion criteria (if applicable), how OS was defined (randomized before or after surgery), their followup period, how many patients completed the planned chemo course, chemo dose density etc . Some extra info/examples are included in these papers: 31229583, 33787841.

Answer: We added percentages for tumor location in table 1, follow-up period in table 2, percentage of patients completed the planned chemo course in table 2.

Margin status was in table 1 (R0 %), chemotherapy dose density was in table 2.

We added the time of randomization in the included studies. Randomization in all 5 studies occurred post-surgery. Line 162-163.

6) In my opinion, there are too many figures and the reader gets lost (especially since the figure legends are not readily available). I would recommend the authors include up to 4-5 figures (choose the most important ones) and submit the rest as supplemental material.

Answer: We reduced the number of figures and relegated the rest to the supplementary materials.

7) The manuscript needs to be reviewed by a native English speaker for syntax /grammar (i.e. use of past tense instead of present tense etc).

Answer: Our manuscript was polished by a native English speaker to enhance its quality.

8) Some of the DOI links in the references are not working, please double check and update accordingly.

Answer: We checked all the DOI links in the references and made sure they were all working.

Reviewer #4: The design and statistical analyses of this meta-analysis are reasonable, and the search process appears to have been conducted effectively. However, I would like to point out two issues that the author should address.

Firstly, it has come to my attention that there is already one paper titled 'meta-analysis in resected biliary tract cancer (the sixth paper in the references). Therefore, it is important for the author to clarify what new contributions this paper will make to the existing literature.

Answer: The meta-analysis conducted by Horgan et al included only 1 randomized trial. Line -. The inclusion criteria in our analysis were RCTs, providing stronger evidence for the adjuvant therapy in resected biliary tract cancer.

Secondly, in the 'Revised Manuscript with Track Change', there should be a "blank space" inserted in front of '3 years' on Line 173.

Answer: We corrected the mistake.

Reviewer #5: The manuscript titled "Adjuvant chemotherapy compared with observation in patients with resected biliary tract cancer: a systematic review and meta-analysis of randomized controlled trials" provides a comprehensive evaluation of the efficacy of adjuvant chemotherapy in patients with resected biliary tract cancer (BTC). Here are critical points and observations:

### Title and Author Information

- The title clearly reflects the content and purpose of the study.

- The authors' affiliations are listed, indicating a multidisciplinary and collaborative effort.

- The corresponding author and contact information are provided.

### Abstract

- The abstract succinctly summarizes the study's objectives, methods, results, and conclusions, which provides a clear overview for readers.

- It includes key findings, such as the impact of adjuvant chemotherapy on relapse-free survival (RFS) and overall survival (OS) in patients with lymph node-positive status.

### Introduction

- The introduction provides necessary background information and context for the study.

- It explains the significance of the clinical issue and the rationale behind conducting the study, which justifies the need for this systematic review and meta-analysis.

### Methods

- Comprehensive databases were searched, with a clear outline of the search strategy, which indicates thoroughness.

- The inclusion and exclusion criteria are well-defined, ensuring the selection of relevant studies.

- The methodological approach, including data extraction and quality assessment, is meticulously detailed.

- The statistical analyses are described, with software and methods mentioned, enhancing reproducibility.

### Results

- The results are thoroughly detailed, providing information on study selection, patient characteristics, and outcomes.

- The heterogeneity among studies has been explained, contributing to transparency.

- Subgroup analyses and their implications are particularly well-detailed, which helps understand nuances in the data.

### Discussion

- The discussion contextualizes the findings within the existing literature.

- Limitations are openly discussed, providing a balanced view of the study's implications.

- Suggestions for future research are valuable and point towards potential advancements in the field.

### General Observations

- The study presents a well-organized flow of information from the background to the implications of the findings.

- References are current and relevant, indicating comprehensive literature engagement.

- Tables and figures are likely informative (though not viewable in this format), providing crucial visualization of data.

- The writing style is scholarly and seems to maintain clinical research formalities.

### Specific Considerations for Improvement

- There may be a need for a more in-depth exploration of bias within included studies, especially given the open-label nature of some trials.

Answer: All the studies included in the analysis were open-label, we discussed it in the limitation part. Line 326-328.

- Consider discussing the geographic distribution of the included studies and its potential impact on the findings.

Answer: We added the geographic distribution of different studies in Table 1 and discussed the potential impact in the limitation part. Line 331-336.

- Ensure a thorough proofread to catch any typographical errors and inconsistencies in formatting (not evident from the provided text).

Answer: We have carefully checked our manuscript and corrected mistakes to enhance its quality.

### Final Thoughts

- The study seems to add valuable information about the efficacy of adjuvant chemotherapy for BTC.

- It highlights the importance of patient selection (lymph node-positive patients) for adjuvant chemotherapy.

- Further research, as recommended by the authors, is imperative to solidify the findings and potentially influence clinical practice.

By addressing these considerations, the paper could make an even more robust contribution to the field and serve as a credible source of information for clinical decision-making in the treatment of resected BTC.

Reviewer #6: Thank you for the opportunity to review your manuscript which evaluated the benefit of adjuvant chemotherapy in patients with post-resected biliary tract cancer. Your meta-analysis included 5 studies with different patient populations, chemotherapy regimens, and different outcome measures. Overall you found significant benefits for adjuvant chemotherapy in patients without R0 resection and those with node-positive disease. Due to the limited evidence in this field, your study makes a significant contribution.

I have a few minor recommendations:

1. The are some grammatical errors that need to be corrected

Answer: We have carefully checked our manuscript and corrected mistakes to enhance its quality.

2. There were few studies in the subgroup analysis, and the overall benefit appeared to be due to the influence of a dominant study (Kohei 2023- Benefit of lymph node-positive disease). You need to be sensitive to this and address it in your discussion.

Answer: We added sensitive analysis in our manuscript. Line 237-246.

---

## [Decision Letter · Decision Letter 2]

3 Apr 2024

Adjuvant chemotherapy compared with observation in patients with resected biliary tract cancer: a systematic review and meta-analysis of randomized controlled trials

PONE-D-23-30589R2

Dear Dr. Wang,

We’re pleased to inform you that your manuscript has been judged scientifically suitable for publication and will be formally accepted for publication once it meets all outstanding technical requirements.

Kind regards,

Jincheng Wang

Academic Editor

PLOS ONE

Additional Editor Comments (optional):

Since authors have fully addressed comments. I think this paper can be accepted.

Reviewers' comments:

Reviewer's Responses to Questions

**Comments to the Author**

1. If the authors have adequately addressed your comments raised in a previous round of review and you feel that this manuscript is now acceptable for publication, you may indicate that here to bypass the “Comments to the Author” section, enter your conflict of interest statement in the “Confidential to Editor” section, and submit your "Accept" recommendation.

Reviewer #3: (No Response)

2. Is the manuscript technically sound, and do the data support the conclusions?

Reviewer #3: (No Response)

3. Has the statistical analysis been performed appropriately and rigorously? 

Reviewer #3: (No Response)

4. Have the authors made all data underlying the findings in their manuscript fully available?

Reviewer #3: (No Response)

5. Is the manuscript presented in an intelligible fashion and written in standard English?

Reviewer #3: (No Response)

6. Review Comments to the Author

Reviewer #3: (No Response)

7. PLOS authors have the option to publish the peer review history of their article (what does this mean? ). If published, this will include your full peer review and any attached files.

**Do you want your identity to be public for this peer review?** For information about this choice, including consent withdrawal, please see our Privacy Policy .

Reviewer #3: No

---

## [Editor Report · Acceptance letter]

PONE-D-23-30589R1

Adjuvant chemotherapy compared with observation in patients with resected biliary tract cancer: a systematic review and meta-analysis of randomized controlled trials

Dear Dr. Wang:

I'm pleased to inform you that your manuscript has been deemed suitable for publication in PLOS ONE. Congratulations! Your manuscript is now with our production department.

Kind regards,

on behalf of

Dr. Alessandro Rizzo

Academic Editor

PLOS ONE